# Preparation of Ultrafine-Grained Continuous Chips by Cryogenic Large Strain Machining

**Haitao Chen, Baoyu Zhang, Jiayang Zhang and Wenjun Deng ***

School of Mechanical and Automotive Engineering, South China University of Technology, Guangzhou 510641, China; 201720100128@mail.scut.edu.cn (H.C.); 201810100060@mail.scut.edu.cn (B.Z.); z.jiayang@mail.scut.edu.cn (J.Z.)
* Correspondence: dengwj@scut.edu.cn; Tel.: +86-20-8711-4634

**Abstract:** Conventional orthogonal machining is an effective severe plastic deformation (SPD) method to fabricate ultrafine-grained (UFG) materials. However, UFG materials produced by room temperature-free machining (RT-FM) are prone to dynamic recovery, which decreases the mechanical properties of UFG materials. In this study, the cryogenic orthogonal machining technique was implemented to fabricate chips that have an abundant UFG microstructure. Solution-treated Al-7075 bulk has been processed in cryogenic temperature (CT) and room temperature (RT) with various machining parameters, respectively. The microstructure, chip morphology and mechanical properties of CT and RT samples have been investigated. CT samples can reach a microhardness of 167.46 Hv, and the hardness of CT samples is higher than that of the corresponding RT samples among all parameters, with an average difference of 5.62 Hv. Piecemeal chip obtained under RT has cracks on its free surface, and elevated temperature aggravates crack growth, whereas all CT samples possess smoother surfaces and continuous shape. CT suppresses dynamic recovery effectively to form a heavier deformation microstructure, and with a higher dislocation density in CT samples, they further improve the chips' hardness. Also, CT inhibits the formation of solute cluster and precipitation to enhance the formability of material, so that continuous chips are formed.

**Keywords:** Al 7075 alloy; free machining; cryogenic; microstructure; mechanical properties

## 1. Introduction

Nanostructured Al 7XXX Series alloys, based on the Al–Zn–Mg–Cu system of nanocrystalline (NC) and ultra-fine grained (UFG) structures, have drawn attention because of their exceptional mechanical properties [1]. In the past decades, severe plastic deformation (SPD) methods, such as accumulative roll bonding (ARB) [2], equal channel angular pressing (ECAP) [3], high-pressure torsion (HPT) [4] and large strain extrusion machining (LSEM) [5–7], have been used for achieving microstructure refinement in metals and alloys. Nevertheless, these SPD methods have a few disadvantages and limitations. For instance, multiple deformations have to process during ECAP to get large plastic strains, the processing efficiency of LSEM is low on account of the fact that the chips are easily blocked in the extrusion tool's channel, ARB and HPT need to consume more energy.

Orthogonal free machining (FM) acts as an efficient SPD method of imposing a large strain to get UFG metals within a single stage of deformation, tool rake the face contact with the chip surface to ensure the chip's geometry can be controlled during chip formation, and varied strain imposed on chips realized by adjusting processing parameters. It is recognized that when the workpiece moving direction is perpendicular to the tool's main cutting edge and the tool's inclination angle equal to 0°, the turning model can be simplified to plane strain (2-D) machining. The 2D orthogonal cutting model and corresponding machining parameters are shown in Figure 1.

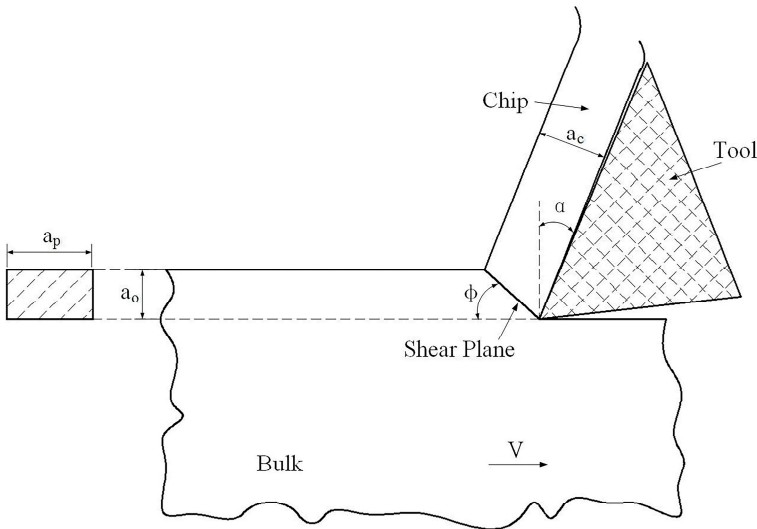

**Figure 1.** Schematic picture of plane strain (2-D) machining.

The shear strain ($\gamma$) imposed on the shear plane is given by [8]:

$$\gamma = \frac{\cos \alpha}{\sin \phi \cos(\phi - \alpha)} \tag{1}$$

where $\alpha$ represents the rake angle, and the shear angle ($\phi$) is calculated from measuring deformed chip's thickness ($a_c$) and undeformed chip's thickness ($a_0$), as in the following equation:

$$\phi = \arctan\left(\frac{\cos \alpha}{\frac{a_c}{a_0} - \sin \alpha}\right) \tag{2}$$

Over the past decade, a lot of scholars applied machining to produce UFG materials. Swaminathan et al. [9] elaborated that plane strain machining is a low-cost and mass production way for making NC and UFG materials. Deng et al. [10] successfully prepared UFG material with an average size of 200 nm by machining carbon steel. However, SPD imposed on the Al 7XXX series alloy at RT gets bad deformability due to the formation of solute clusters and metastable particles, and they result in samples cracking [11]. A series of scholars successively put forward solutions such as processing at elevated temperature or pre-aging treatments before SPD [3,12], but these methods lead to either material softening or formation of a coarse and stable incoherent η phase by over aging. In turn, the strength of the Al 7XXX series alloys decreases. Moreover, the elevated temperature which occurred during SPD at ambient temperature causes material dynamic recovery, recrystallization and grain growth, which sacrifices material mechanical strength [2,13]. Therefore, cryogenic temperature (CT) SPD is developed to conquer the aforementioned limitation. Yu et al. [14] found the ultimate tensile stress of Al/Ti/Al laminate sheets processed by cryogenic roll bonding was 36.7% higher than that by room-temperature roll bonding. By the way, edge cracks also appear in the latter, but the former is in good shape. Panigrahi et al. [15] concluded that producing high strength UFG Al 7075 alloys with high angle grain boundaries can be achieved under cryo-rolling with a strain of 3.4. In addition, Shi et al. [16] employed cryogenic rolling to suppress dynamic recovery and accumulation high-density dislocation in the Al 5052 alloy, which simultaneously raised the yield strength and tensile strength of the alloy.

To date, the investigation of UFG materials' mechanical properties and formability under cryogenic temperature free machining (CT-FM) is limited. It is well known that Al 7XXX series alloys are one of the precipitation-hardenable materials. The principal strengthening mechanisms of Al 7XXX series alloys are grain refinement strengthening, dislocation strengthening and precipitation strengthening, these strengthening mechanisms are closely tied with temperature.

So, it will be meaningful and valuable to study how these factors influence the mechanical properties and formability of materials which are subjected to room temperature free machining (RT-FM) and CT-FM. In the present work, Al 7075 alloy was chosen for research material and processed by RT-FM and CT-FM to get the UFG microstructure. It is well known that mechanical properties and microstructure are closely tied, and we have discussed microstructure evolution and the variation of microhardness versus velocity. Also, Al 7075 is an aging hardenable alloy, second phase precipitate from the matrix influences chip formability, the processing performance is interpreted based on chip thickness and morphology according to the correlated impacts of microstructure, temperature and strain. All in all, this study aims to develop a processing strategy for obtaining UFG Al 7075 alloy and analyze the advantage of CT-FM relative to RT-FM via studying mechanical properties and formability.

## 2. Materials and Methods

The starting commercial Al 7075 alloy was processed into rods with 70 mm in outer diameter, 7 mm in thickness and 20 mm in length, while the chemical compositions of the selected workpiece are shown in Table 1. Subsequently, the workpiece was under solution treatment (ST) at 490 °C for 6 h, and quenched in water immediately later.

**Table 1.** Chemical compositions of the Aluminum 7075 alloy (in wt%).

| Al | Zn | Mg | Cu | Fe | Si | Mn | Cr | Ti |
|---------|------|------|------|------|------|------|------|------|
| Balance | 5.60 | 2.50 | 1.60 | 0.50 | 0.40 | 0.30 | 0.23 | 0.20 |

The ST material was immersed into liquid nitrogen (LN) for at least 20 min until white smog no longer comes out from the LN tank before each processing. As shown in Figure 2a (Figure 2b is an enlarged view of the dotted area in Figure 2a), during cryogenic machining, liquid nitrogen was continuously sprayed from the liquid nitrogen nozzle (LN nozzle) to avert the temperature rising. For comparison, the workpiece is also subjected to room temperature (RT) cutting with the use of cutting fluid. The machining parameters are shown in Table 2. Each set of parameters was repeated at least three times to reduce the experimental errors. RT-FM and CT-FM experiments were conducted on C6140A lathe (Guangzhou Machine Tool Works Co., Ltd., Guangdong, China). To slow down the effect of natural aging, which causes precipitation, and further influence the mechanical properties of Al 7075 chips, the specimen was frozen in the fridge with a temperature below −35 °C.

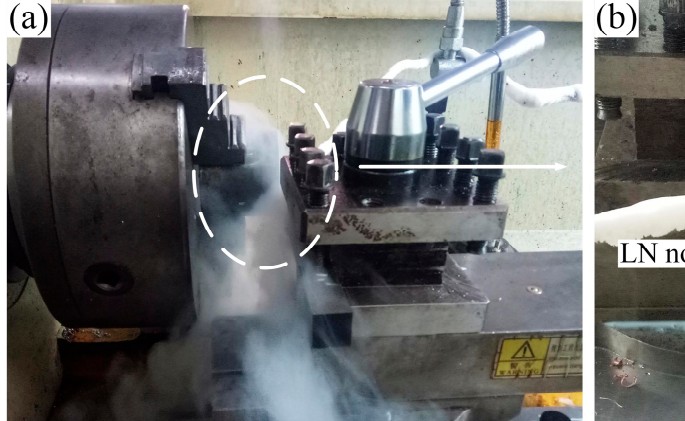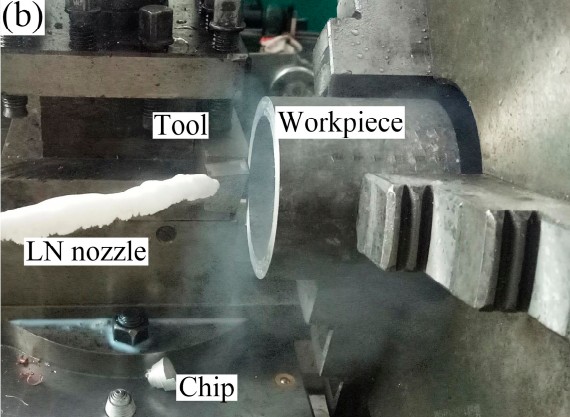

**Figure 2.** Schematic view of cryogenic temperature free machining (CT-FM): (**a**) Overall view; (**b**) view of the dotted area in Figure 2a (taken from a reverse angle).

**Table 2.** Machining parameters.

| Variable Parameters | | | |
|---|---|---|---|
| **Rake Angle (°)** | **Cutting Velocities(mm/s)** | | |
| 10 | 92 | 183 | 385 |
| 15 | 476 | 660 | 770 |
| 20 | 9553 | 1319 | 2053 |
| **Fixed parameters** | | | |
| $a_P$ | | $a_0$ | |
| 7mm | | 0.5mm | |

The Vickers microhardness measurements were conducted on the chip's plane, which contacts with the tool rake face. The indenter applies, using 100 g loads on the chip's surface and maintaining pressure for 15 s. The survey was conducted on random areas of the samples' surface, and at least 15 values were selected to calculate the average hardness of each chip. The microstructure of the ST Al 7075 alloy was surveyed by an optical microscope (OM). The sample preparation process is as follows: firstly, making chips into metallographic specimens; secondly, mechanically polished by abrasive papers with 600, 1000, 1500, 2000 grit in order; thirdly, fine polishing was carried on samples with diamond paste of 1 μm to achieve a mirror finish; finally, the specimens were etched by Keller's reagent (95 mL $H_2O$, 2.5 mL $HNO_3$, 1.5 mL HCl, and 1.0 mL HF). JEM-1400 PLUS transmission electron microscopy (Japan Electronics Co., Ltd., Tokyo, Japan), which operates at an acceleration voltage of 200 kv, was used to characterize the microstructure and precipitation of the UFG chips. Transmission electron microscopy (TEM) samples were first mechanically ground to achieve a thickness of 50 μm, then punch samples to get a foil with a diameter of 3 mm, finally conduct electrolytic double jet (10 vol% $HCLO_4$ in ethanol, 20 V, −25 °C) and argon ion beam on foils.

## 3. Results and Discussion

### 3.1. Produced Chips

Chip formation is a complicated procedure, which is determined by synthesizing the effect of material properties and machining parameters, workpiece experiencing different shear strain and deformation heat result in different chip thickness and morphology. The morphology of produced chips at different machining speeds with a 10° rake angle is expressed in Figure 3, the chips' shape, which was produced at low velocities is discontinuous and curly, due to the chip formation mechanism being in the Built-Up Edge (BUE) mode. During the low speed (V < 476 mm/s) cutting process, the chip's inner face which contacts with the tool rake face is subjected to a force due to the advancement of cutter and material softening that causes it to bend away from the rake face. Subsequently, self-contact happens on the chip's free side. After a certain period, the chip has been subjected to excessive force and causes fracturing. Machining aluminum 7075 alloy at room temperature is easy to produce serrated chips, whereas the corresponding CT chip has a more complete morphology. Trent et al. [17] stated that during the cutting process, when velocities exceed a threshold velocity value, strip continuous chip emerged because the chip formation mechanism has changed into a seizure mechanism. As the cutting speed increased to approximately 953 mm/s, continuous ribbon and enlarge bending radius chips which embody flatter morphology were formed under CT, but pronounced crack and fragile block chips appear under RT. The thickness of chips (Figure 4) reflects the deformation behavior of materials from the side. Generally speaking, the thickness of chips at room temperature is larger than that at low temperature. The larger the cutting speed, the more obvious the phenomenon of the serrated chip which was processed in RT-FM appears, and the thickness continues to increase, where the chip thickness achieves 1.13 mm at 2053 mm/s. However, the higher the speed, the better the production of continuous ribbon chips in CT-FM.

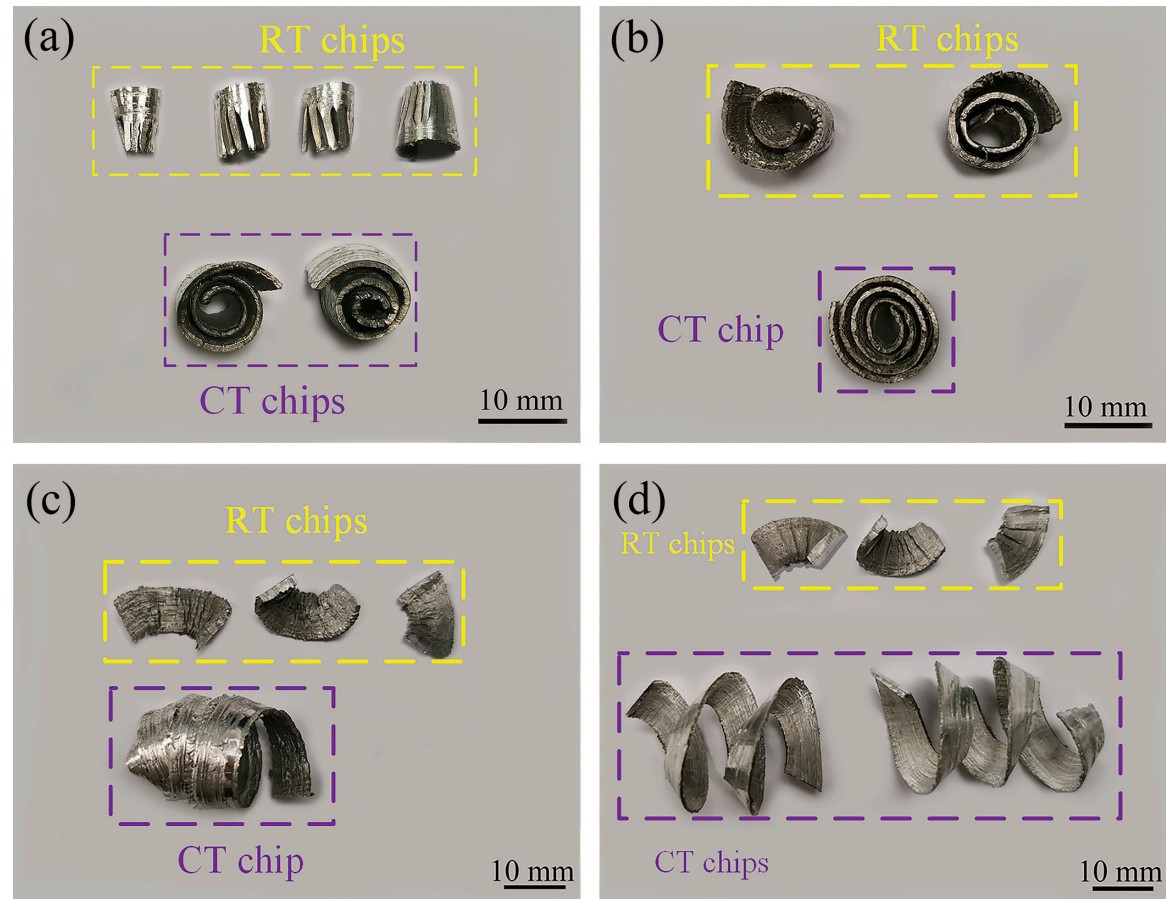

**Figure 3.** CT-FM and room temperature free machining (RT-FM) chip morphology at different cutting speeds at 10° rake angle: (**a**) 92 mm/s; (**b**) 476 mm/s; (**c**) 953 mm/s; (**d**) 2053 mm/s.

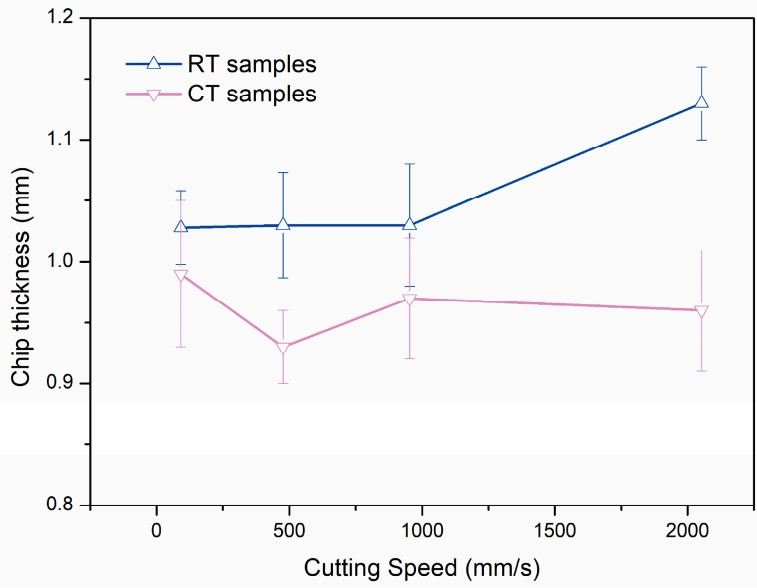

**Figure 4.** Ultrafine-grained (UFG) chip thickness versus cutting velocity under 10° rake angle.

It is reported [18] that when machining aluminum alloys, the stiffness of the chip area in contact with the rake face of the tool is reduced, which causes a reduction in the stress at the tool tip. When chips flow out from the prime deformation zone (PDZ), intense friction between chips and the rake face will make the temperature in the second deformation zone (SDZ) higher than the chip's free side.

As a result, the equivalent plastic strain and temperature around the chip-tool's rake face interface will spread to the free side of the chip, which indicates that shear rupture happens in PDZ. Consequently, in the process of workpieces flowing out from PDZ and turning into chip, materials tend to deform along the direction perpendicular to the chip-tool interface, leading to increasing chip thickness. Along with this, during RT-FM, the heat of deformation causes the solute atom cluster (or GP zone) and other phases (such as η′ or η) precipitated in the matrix to interact with the deformation dislocation, further causing chip hardening and decreasing ductility, leading to chip breaking eventually. Under the combined influence of stress, strain, temperature and precipitates, localized adiabatic shear takes place on the RT samples. However, during the CT deformation process, the solute diffusion is hindered, so that the precipitation of the strengthening phase is inhibited, the movement of dislocation is not prevented, so the dislocations are continuously generated, moved and entangled to form a characteristic deformation structure, and the plastic deformation of the alloy is continuously performed, thereby significantly reducing the occurrence of the shear instability zone. Moreover, aluminum alloy is a kind of face-centered cubic (fcc) lattice materials, even though under the low-temperature conditions, fcc materials' impact values are still high, and its plasticity remains unchanged or changed slightly, and that means without the phenomenon of low-temperature brittleness. Besides, the workpiece will improve the work hardening ability under low-temperature deformation [19]. The formation and accumulation of high dislocations density in the CT deformation process of aluminum alloy with dislocation slip improves the plasticity of the alloy under ultra-low temperature conditions, so that the chip can obtain uniform and stable plastic deformation. This testifies as to the result that the RT chip thickness is larger than the CT chip thickness.

### 3.2. Hardness

The mechanical properties of the chip are affected by multiple factors, whereas the main effect parameters are the large strain, high strain rate and the heat produced during machining. The initial microhardness of the solution-treated Al 7075 is 98 Hv. Based on existing knowledge, four strengthening mechanisms can improve the mechanical properties of aluminum alloys: (a) solution strengthening, (b) dislocation strengthening, (c) precipitation strengthening, (d) grain refinement strengthening. When Al 7075 alloy materials are conducting solution treatment and quenched in water immediately to form a supersaturated solid solution, and solute atoms like Zn, Mg, Cu, Fe and Mn are dissolved into Al matrix, the solute and parent atoms' discrepancy in sizes, valency and modulus will strengthen the alloys' mechanical properties [15]. After experiencing a certain extent of deformation, the chips' grain size has been greatly refined, according to Hall–Petch equation, grain size is inversely proportional to microhardness, and the hardness of the CT-FM and RT-FM samples at a rake angle of 10° and a speed of 92 mm/s has increased from 98 to 167 Hv (nearly 70.4% increase) and 98 to 166 Hv (nearly 68.6% increase), respectively.

$$H_V = H_0 + kd^{\frac{-1}{2}} \tag{3}$$

In this formula, $H_V$ represents the chip's hardness, $H_0$ and k are both materials constants, and d is the average grain size of ultrafine-grained materials.

During the RT-FM, the temperature in the chips which separates from the workpiece is reduced to room temperature by cutting fluid, the cooling time is very short, but precipitation phenomena are still likely to occur even during CT-FM. These coherent nanosized precipitates play an impeding dislocation motion role. In comparison to RT-FM, chips under the CT-FM condition will accumulate more dislocation densities, wherein the high dislocation densities can improve the ability to resist the deformations of materials by decreasing the average dislocation movement's free path [20].

Figure 5 displays UFG chip microhardness and shear strain versus varied cutting speeds and rake angles. In general, microhardness changes in a range of 146 to 167 Hv, and shear strain, which is calculated from Equation (1), ranges from 1.6 to 2.4 with different samples. As can be seen from the image, ultrafine-grained chips' hardness diminishes from the maximum to a stable value vs.

cutting velocity. The hardness of CT samples is larger than the hardness of RT-FM samples among all corresponding processing parameters, and the mean difference in hardness between CT and RT samples in all parameters is 5.62 Hv. The maximum difference in hardness between CT and RT samples is up to 11.54 Hv at a rake angle of 20° and a speed of 770 mm/s, whereas the minimum worth attains 0.09 Hv at a rake angle of 20° and a speed of 92 mm/s. Kanani et al. [21] revealed that if machining velocities are lower than the threshold value which converts the chip forming mechanism, the actual shear strain imposed on the chip is smaller than the strain calculated by Equation (1). Moreover, the generated heat at low machining velocities might not have a considerable magnitude. The factor that affects chip hardness at low speed is low strain and a trace amount of heat, thus the difference between the hardness of CT chips and the hardness of normal RT chips is relatively small. The PDZ and SDZ's temperature enhances along with the increasing velocities, so the RT chip's microstructure is prone to recrystallize and annihilate dislocations. But cryogenic cutting can suppress dynamic softening, because under CT conditions, the dynamic recovery is prevented effectively, and workpieces can store more dislocation densities, and thus the difference in hardness between CT-FM and RT-FM samples will enlarge as speed increases. It can be discovered that when the cutting velocity is higher than 476 mm/s, the hardness will suddenly decrease by 6 to 11 Hv, except for the processing parameters at the rake angle of 20° at cryogenic temperature. That may be because CT under this parameter has played an exceptional cooling role, or material deformation has reached a limited value. It can be concluded that specimens with minimal machining velocities have the maximum hardness. When chips are imposed to shear strain, its hardness will enhance because of working hardening. Shear strain magnifies with raising velocity, nevertheless, chip microhardness begins to descend.

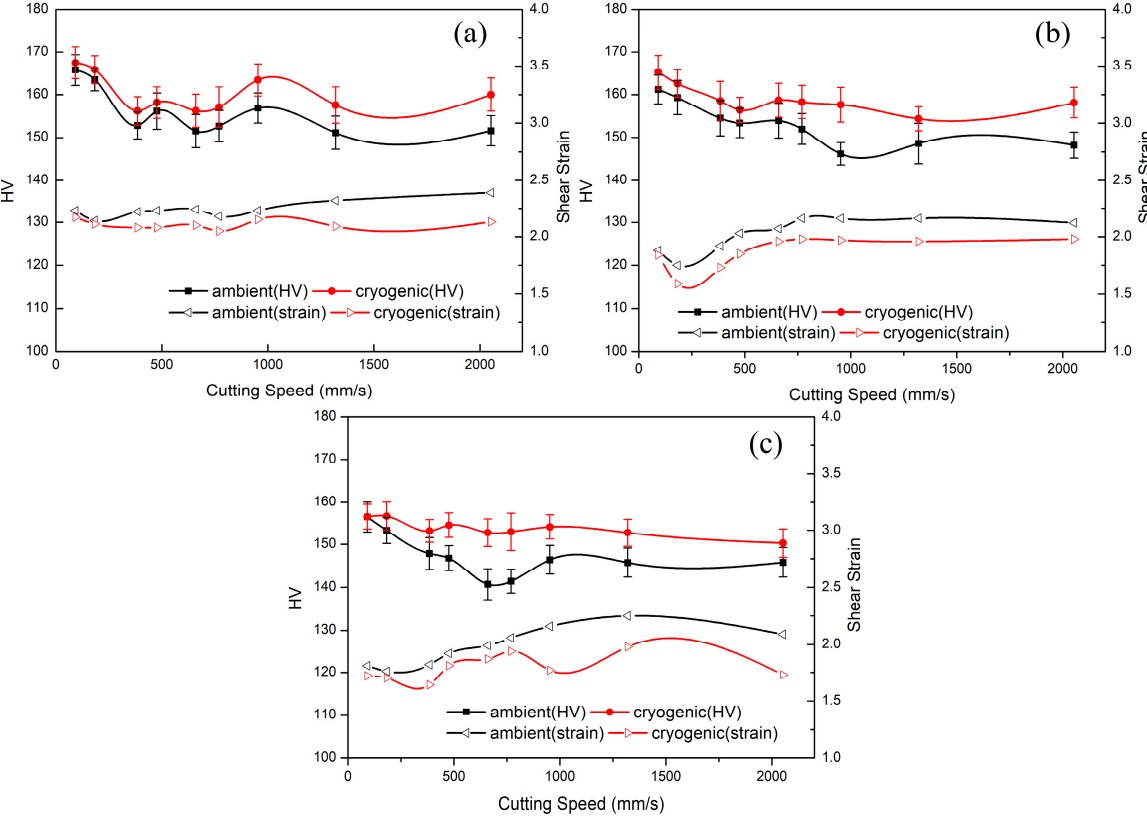

**Figure 5.** Vickers hardness and shear strain of the C-FM and RT-FM Al 7075 chips vs. the tool's rake angle at: (**a**) 10°; (**b**) 15°; (**c**) 20°.

Generated heat (which comes from two resources: one is for materials deformation, and other is for friction between the tool rake face and the chip) will also increase with increasing speed, it can

activate and motivate the softening mechanism, so that it will counteract the effect of shear strain on materials strengthening. When the former effect is superior to that of the latter, chips' hardness descends with increasing speed. Chip hardness is stable within a certain range finally, and it can be deduced that when machining velocities exceed a certain value, the interaction between material recovery and strengthening factors will reach a balanced state.

### 3.3. Microstructure

The mechanical properties of materials are closely related to their microstructures, where OM and TEM is used to characterize this phenomenon. It has chosen ST samples for OM characterization and UFG samples for TEM characterization. The grain size was calculated by the linear intercept method and measured on the Dark-field (DF) images (Figure 6), where bright areas are used to identify the grain, and the average amount of measuring grains is more than 100, and the grain statistical distributions are indicated in Figure 7. Besides, for some elongated grains in the chips' microstructure, the mean width (along a small dimension) of grain is regarded as the grain size [22].

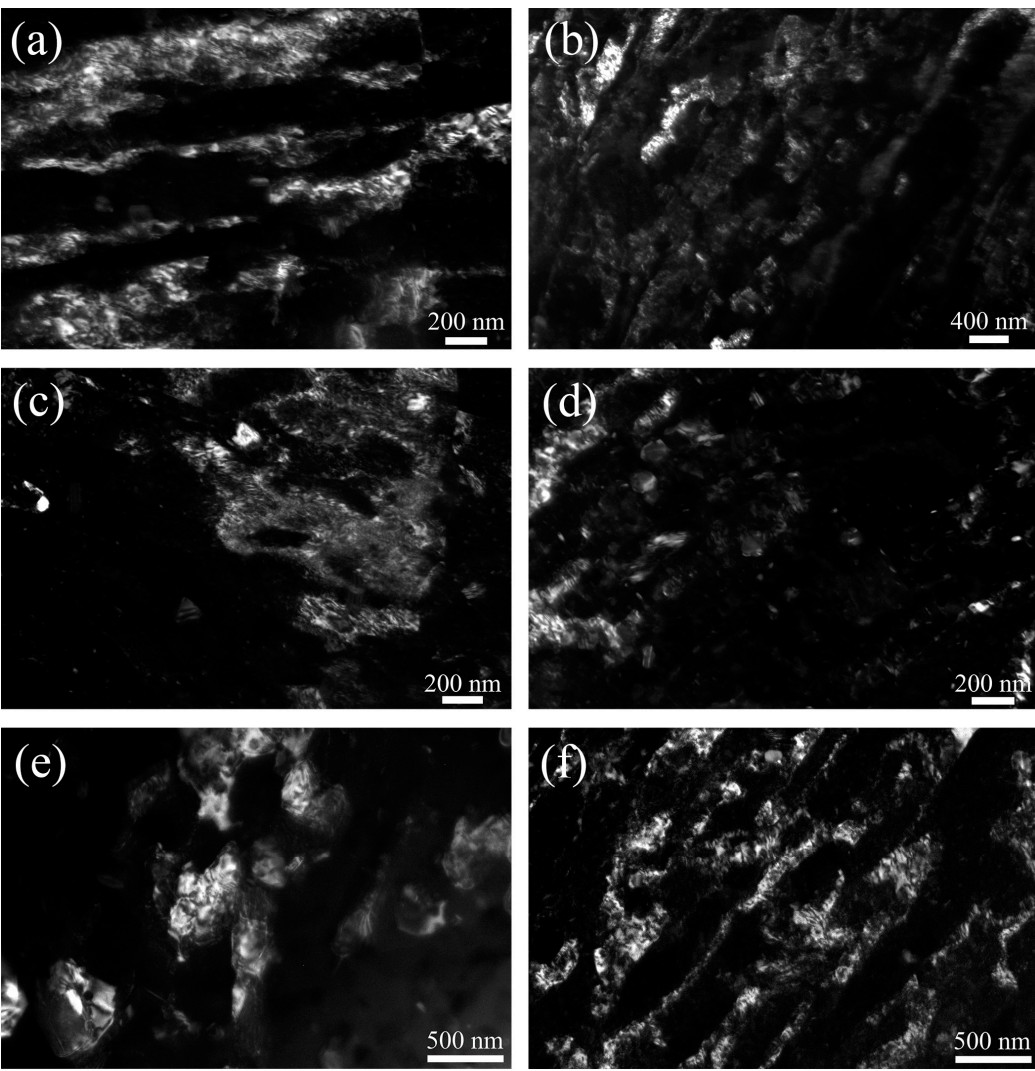

**Figure 6.** Dark-field (DF) TEM microstructure of produced UFG chips under different cutting conditions: (**a**) RT samples at $\alpha$ = 15° and V = 476 mm/s, (**b**) CT samples at $\alpha$ = 15° and V = 476 mm/s, (**c**) RT samples at $\alpha$ = 15° and V = 953 mm/s, (**d**) CT samples at $\alpha$ = 15° and V = 953 mm/s, (**e**) RT samples at $\alpha$ = 20° and V = 2053 mm/s, (**f**) CT samples at $\alpha$ = 20° and V = 2053 mm/s.

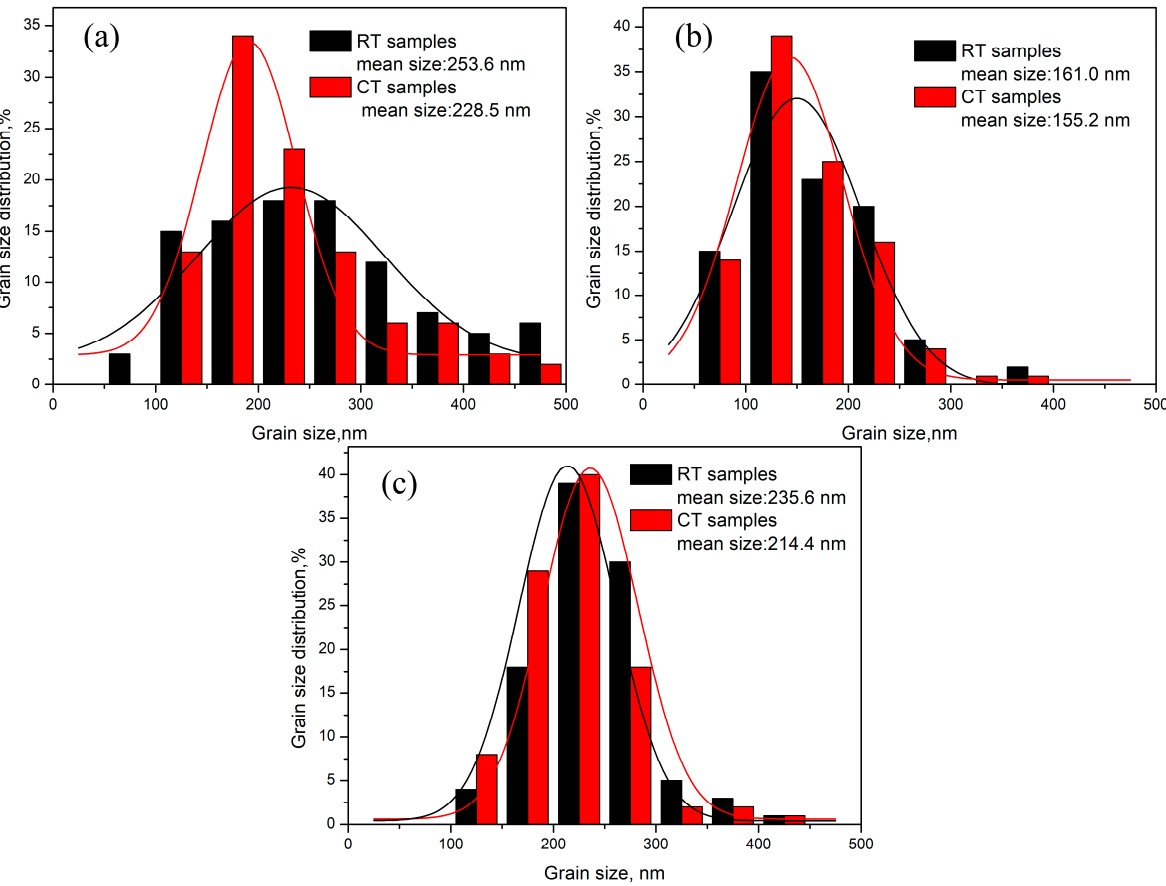

**Figure 7.** Grain size distribution of UFG samples: (**a**) α = 15° and V = 476 mm/s; (**b**) α = 15° and V = 953 mm/s; (**c**) α = 20° and V = 2053 mm/s.

The optical image of as-received material after solution treatment is shown in Figure 8. After 6 h treatment at 490 °C, the sample's microstructure exhibits clear and large size grain boundaries. Nevertheless, the Al matrix exists in the insoluble phase, such as the intermetallic $Al_7Cu_2$Fe phase and S phase ($Al_2$CuMg) particles, and coarse black particles (as shown by the green arrow), and then these black particles may be $Mg_2$Si, which forms in temperature around 490 °C. Furthermore, coarse $Mg_2$Si particles are detrimental to the strength and hardness of the alloy, because they hinder the dissolution of the solute element (such as Zn, Mg, Cu) into the matrix [23]. The raw Al-7075 alloy did not experience severe deformation, and a number of strengthening phases dissolve into the matrix, so dislocation strengthening, grain refinement strengthening and precipitation strengthening did not exist, and the only strengthening mechanism is solution strengthening. To improve the material's mechanical properties, it is necessary to apply SPD.

TEM microstructures of different samples with Bright-field (BF) images and selected area electron diffraction (SAED) are illustrated in Figure 9. As for RT-FM samples, it can be detected from Figure 9a that the obtained chip has a nanostructure with elongated grain through shear deformation direction due to flow stress and strain, the original grain boundaries and grains are relatively clear and the fibrosis is serious. The SAED pattern reveals discrete dots, which implies that the observed grain boundaries are the Low Angle Grain Boundary (LAGB). There exists some dislocation tangling and a lot of precipitation with rod-shaped and spherical shapes in this area, the latter is mainly a fine and dense distribution of the GP zone, metastable η′ phase and equilibrium η phase [5]. Among these second phases, the GP zone and η′ phase are major strengthening phases, they not only act as obstacles to dislocation movement, such as the pin grain boundary and dislocation to improve strength, but also affect substances to sustain thermal stability by retarding grain boundaries migration.

Besides, it is also observed as to some good defined extended equiaxed grains, which size differs from 113 to 400 nm. In fact, microstructure evolution during the machining process contain two separate procedures: on the one hand, the chip's microstructure is affected by localized strain, and produced heat simultaneously, which happens in the deformation zone (as dynamic events); on the other hand, when issuing out of there, the heat generated by the former will influence the microstructure (as static events). Although the process is very short, recrystallization could take place due to temperature over 0.5 $T_m$ and an unusually high prior strain results in the rotation of the subgrain; a similar phenomenon was discovered by Campbell et al. [24]. With increasing velocities, the increasing of shear strain aggravates chip deformation, the average grain size has reduced to 161 nm, but the accompanying heat also ascends. Figure 9c shows partial recrystallization (marked by the red arrow) and clear subgrain boundaries. The selected sample deforms under strain and temperature; the original grain was separated into a few subgrains by LAGB. Owing to unequal deformation, dislocation densities were different in vast regions, dislocation climb and cross-slip to rearrange subgrain boundaries, this procedure was usually called "repeated polygonization" [25]. As shear strain rises further, the original grain boundaries will serrate and fracture to form a more distinct boundary. In addition, chip under a higher velocity underwent higher heat, which boosts the progress of recrystallization, so it could contribute to softening mechanism, and hence reduce microhardness. As showed in Figure 9e, it is observed concerning a large number of equiaxed recrystallization grains (as indicated by the blue arrow) and better-defined enlarged subgrain boundaries, where the average grain size has increased to 235.6 nm. Al 7075 alloy is a kind of high stacking fault energy material, where subgrain boundaries absorb dislocation continuously and rotate for potential continuous dynamic recrystallization during hot deformation. In this process the misorientation angle augments gradually, and LAGB transforms to HAGB at the end. Precipitates have grown, coarsening and dislocation nearly disappear, compared to that in Figure 9a,c; recrystallization and grain growth intensify due to coarse and sparse particles weakening the Zener pinning effect. These phenomena have led to the effect of dislocation strengthening, wherein grain refine strengthening and precipitation strengthening fade. Consequently, ultrafine-grained chip hardness reduces, as depicted in Figure 5.

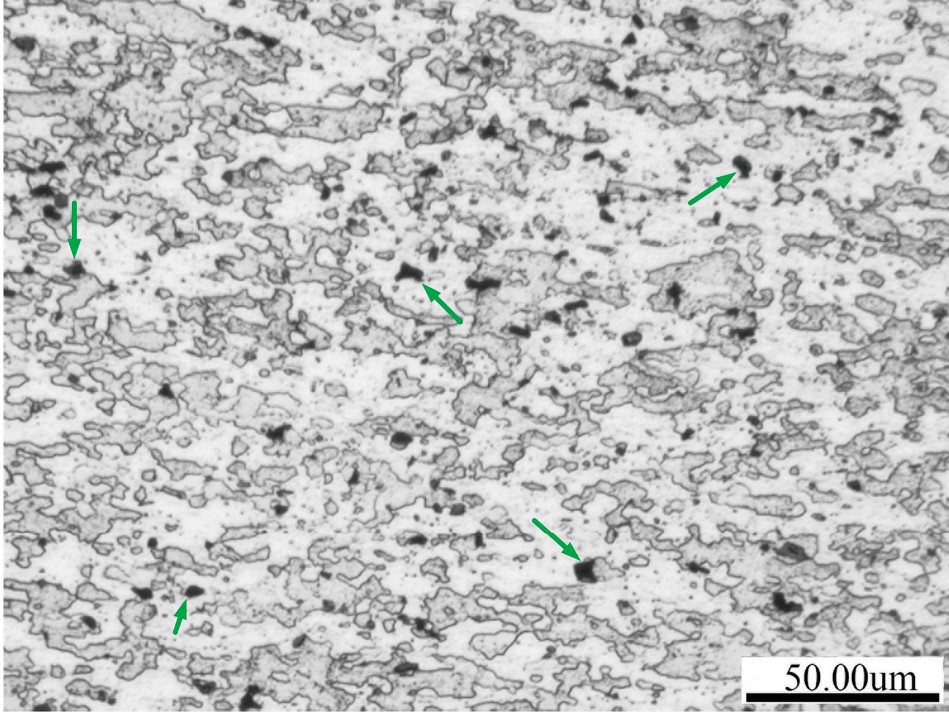

**Figure 8.** Optical microscope (OM) image of solution-treated Al 7075 alloy.

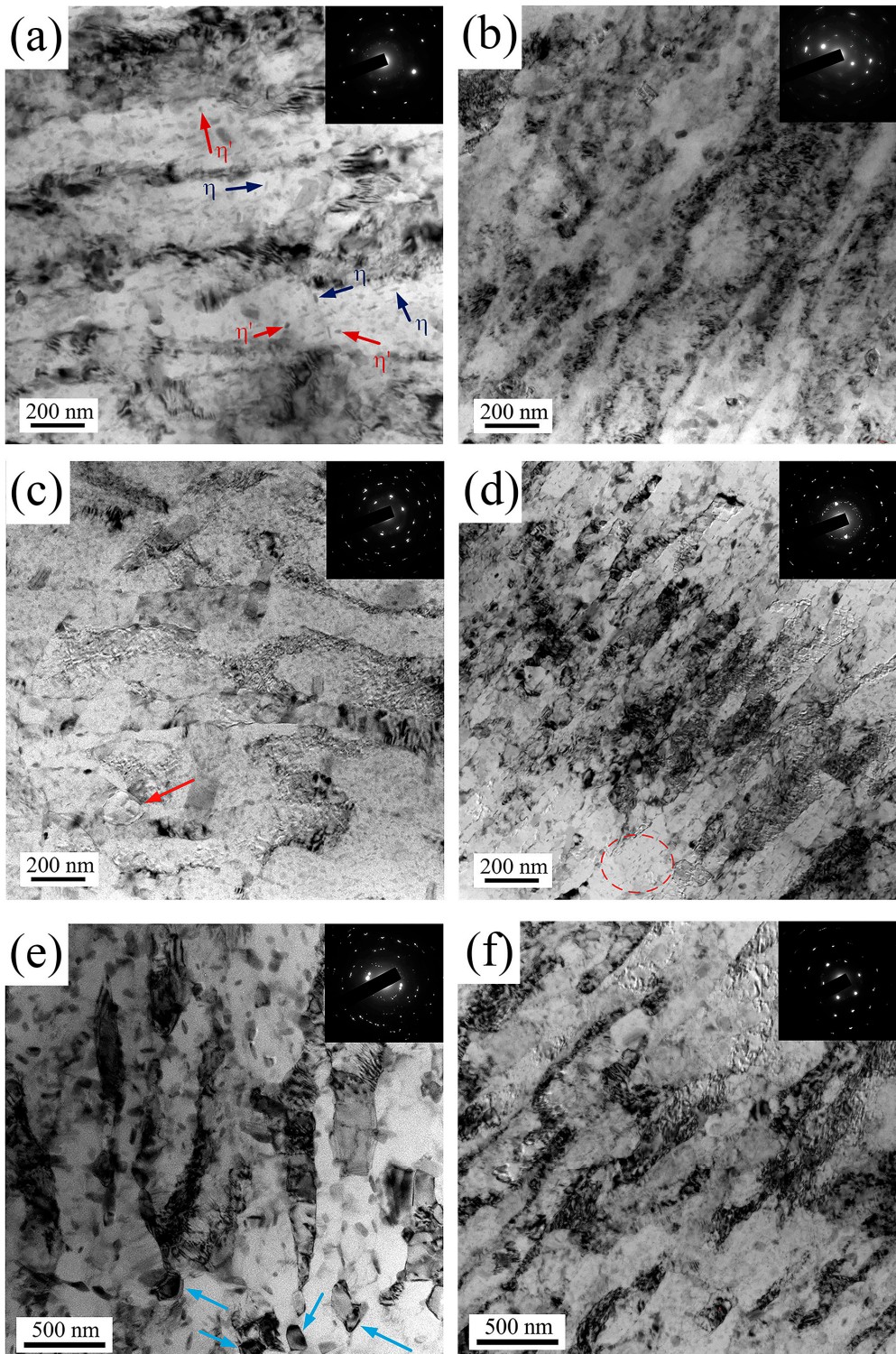

**Figure 9.** Bright-field (BF) transmission electron microscopy (TEM) microstructures of produced UFG chips under different cutting conditions: (**a**) RT samples at $\alpha = 15°$ and V = 476 mm/s; (**b**) CT samples at $\alpha = 15°$ and V = 476 mm/s; (**c**) RT samples at $\alpha = 15°$ and V = 953 mm/s; (**d**) CT samples at $\alpha = 15°$ and V = 953 mm/s; (**e**) RT samples at $\alpha = 20°$ and V = 2053 mm/s; (**f**) CT samples at $\alpha = 20°$ and V = 2053 mm/s.

Whereas heavier deformation structures were observed in corresponding CT-FM samples. Overall, CT samples mean grain size is smaller than RT samples mean grain size, and the distribution of grain

size in CT samples at 100–250 nm is more than that in RT samples. The SAED pattern of Figure 9b demonstrates continuous ring type; it indicates that the microstructure consists of a High Angle Grain Boundary (HAGB). It observes a large number of dislocation tangles in many regions. Furthermore, they lead to many submicron structures, such as dislocation walls and dislocation cells, and the grain size varies in the range of 133 to 460 nm. Although cryogenics can suppress dynamic recovery to a large extent, relatively few precipitations occur in CT-FM chips (Figure 9d, marked by the red, elliptical, dotted frame), which perhaps the CT chip happens through natural aging or strain-induced precipitation during deformation. Figure 9f shows diffused, nonequilibrium and ill-defined grain boundaries which are similar to Figure 9b,d. It is obvious that the dislocation density of the CT-FM chip is higher than that of the RT-FM chip, and the mechanism for this phenomenon is that the low temperature inhibits dynamic recovery effectively, and thus accumulates a higher amount of dislocation density. During RT-FM, the friction between the tool rake face and chip in SDZ and deformation in PDZ will bring relatively high temperature, then appears cross slip of screw dislocation and climb of edge dislocation, which cause an opposite sign of dislocations annihilation and rearrangement, finally dislocation density decreases.

Low temperature inhibits atom diffusing and dynamic recovery, which ensures defects such as dislocation, and vacancy could be saved and accumulated continuously to form a remarkable substructure. Lastly, a severe deformation of the microstructure is obtained. The change in grain size of CT samples versus speed is similar to that of RT samples. However, CT can preserve dislocation density as much as possible, and the variation of grain size is smaller, therefore the fluctuation of CT chips' hardness is less sensitive to speed.

As mentioned before, the deformed aluminum 7075 alloy's mechanical properties are determined by the comprehensive impact of grain size, dislocation and precipitation; the overall strength can be estimated by the following equation [26]:

$$\sigma_y = \sigma_0 + kd^{\frac{-1}{2}} + M\alpha Gb\sqrt{\rho} + \frac{MGb}{1-2r} \tag{4}$$

Where $\sigma_0$, k, d, M, $\alpha$, G, b, $\rho$, r and l represent friction stress, the Hall–Petch constant, grain size, Taylor factor, a constant, shear modulus, Burgers vector, dislocation density, average size of precipitates and precipitates spacing, respectively. As for the second term of Equation (4), the yield strength is inversely proportional to $d^{\frac{1}{2}}$, the chip's strength after FM is enhanced due to imposed strain lead to grain refinement. From Figure 7 it can be concluded that the specimen's grain size at CT-FM is smaller than that at RT-FM, so the grain refinement strengthening influence on CT samples is larger than RT samples. From the third term in Equation (4) it can be found that dislocation density and yield strength are positively correlated, and the factor of forming dislocation consists of interaction between dislocations, second phase, grain boundaries, and temperature. The higher dislocation density in CT samples makes it that the dislocation strengthening effect of CT samples is stronger than the RT samples. During RT-FM, lots of second phases precipitate from the matrix and grow. Owing to this, the particle spacing (l) in the RT samples is smaller than as compared to that of the CT samples, and the average precipitate size (r) in the RT samples is larger than that of the CT samples. According to the fourth term from Equation (4), the precipitate strengthening contribution to RT samples is greater than that of CT samples. But grain-boundary strengthening is the predominant mechanism in the UFG 7075 material [27], the degree of precipitation strengthening in RT samples is not enough to make up for inferior grain refinement strengthening and dislocation strengthening of RT samples, so CT samples' strength is higher than RT samples' strength.

The relationship between yield strength and microhardness is linearly positively correlated [28], and this verifies the higher hardness of CT samples.

## 4. Conclusions

In the present study, the solution-treated Al 7075 was subjected to RT-FM and CT-FM via different combinations of rake angle and velocity. Mechanical properties and microstructure of FM samples have been studied in depth. The following conclusions are made:

1.  The chip's hardness decreases from the maximum value and stabilizes in a certain range with increasing velocity. The CT chip shows higher microhardness, and the difference in hardness between CT chips and RT chips increase with increasing speed.
2.  The grain size of RT and CT samples are both mainly distributed in the range of 100–300 nm. RT samples' microstructures are more greatly influenced by machining parameters, elongated grains, and many precipitates exist in RT chips at the 10°rake angle and 476 mm/s cutting speed. Furthermore, the microstructure at the 20° rake angle and the 2053 mm/s velocity reveal that most of the dislocations are annihilated and accompanied by growing dispersoids and a large number of recrystallizations. However, CT samples remain at a higher dislocation density and smaller grain size among various machining parameters which improve its mechanical properties.
3.  Due to the formation of lots of precipitations in RT-FM, the chip is prone to crack localization and fracture, and especially at high speed, the chip's crack is pronounced, so that sawtooth chips are very likely to be produced in RT-FM. Deformation at CT can quickly take away a lot of heat and delay precipitation and diffusion, improving the dislocation accumulation capability and enhancing the alloy's work hardening rate to ensure uniform and stable plastic deformation of the aluminum alloy, thus producing UFG materials having a smoother and continuous morphology.

**Author Contributions:** H.C. designed work plans, executed all experiments and wrote the article; B.Z. helped to build the research platform and optimize the experiment scheme; J.Z. raised several useful pieces of advice on the structure and picture-making of the paper; W.D. conceived the present work and corrected the manuscript. All authors have read and agreed to the published version of the manuscript.

**Funding:** This research was funded by the National Science Foundation of China (51375174), Fundamental Research Funds for Central Universities (2017ZD024), and Natural Science Foundation of Guangdong Province (S2013050014163, 2017A030313260).

**Acknowledgments:** This work was financially supported by the National Natural Science Foundation of China (51375174), Fundamental Research Funds for Central Universities (2017ZD024), and the Natural Science Foundation of Guangdong Province (s2013050014163, 2017A030313260).

**Conflicts of Interest:** The authors declare no conflict of interest.

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
