# Peer review of "Preparation of Ultrafine-Grained Continuous Chips by Cryogenic Large Strain Machining"

_metals, doi:10.3390/met10030398_

Round 1

Reviewer 1 Report

In this study, the authors investigated the effects of temperature and other machining conditions on UFG formation in Al 7XXX alloy.

The experimental results seem reasonable, and the manuscript is well-prepared.

This article could be published after minor corrections as suggested below.

  1. Page 3 Line 105: “Mosaic specimens” could not be imagined easily. This would be replaced by other adequate and scientific terms.
  2. Page 4 Line 108: “teched” seems to be miss-typing of “etched”.
  3. The authors explain that the machining is an efficient SPD method to obtain UFG metals. However, it seems difficult to imagine the application of the small chips produced by machining. If this study focuses on basic mechanism of UFG formation by SPD, that would be emphasized in the introduction.

Author Response

Dear Editor and reviewers:

Thanks very much for your help and the reviewers’ comments concerning our manuscript which entitled “Preparation of ultrafine-grained continuous chips by cryogenic large strain machining”. Those comments are all valuable and helpful for revising and improving our paper, as well as the significant guiding to our researchers. We have studied comments carefully and made revisions, these revisions are highlighted through the whole manuscript. Also, we have answered reviewers’ comments and explained the detail revisions in manuscript point by point. Now, we hope our revisions can meet with approval.

Response to reviewers’ comments

In this study, the authors investigated the effects of temperature and other machining conditions on UFG formation in Al 7XXX alloy.

The experimental results seem reasonable, and the manuscript is well-prepared.

This article could be published after minor corrections as suggested below.

  1. Page 3 Line 105: “Mosaic specimens” could not be imagined easily. This would be replaced by other adequate and scientific terms.

Thank you so much for your comments. Your comments are very valuable to us. The picture of “Mosaic specimens” is shown in Figure 1. After careful consideration, we think the appropriate name for the OM sample is “metallographic specimens”. We changed “Mosaic specimens” to “metallographic specimens”, the line number of the modified contents is 110.

Figure1. metallographic specimens

Based on your valuable comments, we have modified the manuscript as following:

firstly, making chips into metallographic specimens;

Now we think readers can better understand our paper, please consider our revisions.

  1. Page 4 Line 108: “teched” seems to be miss-typing of “etched”.

Thank you so much for your comments. Your comments are very valuable to us. Due to our carelessness, we made this mistake. We changed “teched” to “etched”, the line number of the modified contents is 112.

Based on your valuable comments, we have modified the manuscript as following:

finally, the specimens were etched by Keller’s reagent (95ml O, 2.5ml , 1.5ml HCl, and 1.0ml HF).

Now we think readers can better understand our paper, please consider our revisions.

  1. The authors explain that the machining is an efficient SPD method to obtain UFG metals. However, it seems difficult to imagine the application of the small chips produced by machining. If this study focuses on basic mechanism of UFG formation by SPD, that would be emphasized in the introduction.

Thank you so much for your comments. Your comments are very valuable to us. Generally speaking, the chips produced by machining possess high mechanical properties, but the size of them is the biggest limitation that confine their industrial application. In this article, we combine cryogenic and machining to develop a new strategy for manufacturing higher mechanical properties and better morphology chips. We aim to study how CT exerts influences on chips’ microstructure evolution and improve chips’ mechanical properties and formability.

Moreover, we have been working on the practical application of UFG chip by increasing material’s size and optimizing tool’s structure. The UFG chips have also been applied in some fields. Conventional micromachining was used to produce high strength UFG materials, and small-scale gear was manufactured from UFG chips as shown in Figure 2(doi:10.1016/j.msea.2008.02.056). The UFG chips can also be used to produce nano-composite materials, the scholar milled the chips into powders and consolidated them with nano-particles(doi:10.1016/j.powtec.2013.07.028).

Figure 2. Micro gear produced by UFG chips

Figure 3. Chips morphology and milled powders

Overall, UFG chips have lots of potential applications, and our next work including applying them to industry. Considering our main focus on this paper is studying the mechanism on how CT affects chips’ mechanical properties and formability, we have added sentences to stress this point, the line numbers of the modified contents are from 71 to75.

Based on your valuable comments, we have modified the manuscript as following:

It’s well known that Al 7XXX series alloys are one of the precipitation-hardenable materials. The principal strengthening mechanisms of Al 7XXX series alloys are grain refinement strengthening, dislocation strengthening, and precipitation strengthening, these strengthening mechanisms are closely tied with temperature. So, it will be meaningful and valuable and to study how these factors influence the mechanical properties and formability of materials which subjected to room temperature free machining (RT-FM) and CT-FM.

Now we think readers can better understand our paper, please consider our revisions.

Reviewer 2 Report

English readability is quite low, and there are numerous mistakes of English grammar. It is very hard for the readers to understand his paper in this form.

Author Response

Dear Editor and reviewers:

Thanks very much for your help and the reviewers’ comments concerning our manuscript which entitled “Preparation of ultrafine-grained continuous chips by cryogenic large strain machining”. Those comments are all valuable and helpful for revising and improving our paper, as well as the significant guiding to our researchers. We have studied comments carefully and made revisions, these revisions are highlighted through the whole manuscript. Also, we have answered reviewers’ comments and explained the detail revisions in manuscript point by point. Now, we hope our revisions can meet with approval.

Response to reviewers’ comments

English readability is quite low, and there are numerous mistakes of English grammar. It is very hard for the readers to understand his paper in this form.

We feel sorry for our manuscript existing lots of grammar mistakes, and we admit that readers may be confused when reading this article. The paper has been carefully revised by a native English speaker to improve readability. We checked every sentence of this manuscript, and corrected every error as much as we can. At the same time, we try not to make the sentences ambiguous or obscure by optimizing sentence structure and replacing some easy-to-understand synonyms. We hope that it would be easier for readers to understand the revised paper.

Reviewer 3 Report

The authors study the influence of various machining parameters on a microstructure evolution and microhardness of the ultrafine grained Al 7075 alloy, processed by orthogonal free machining technique at room temperature (RT-FM ) and at cryogenic temperature (CT-FM). The analysis of advantages of CT-FM relative to RT-FM via mechanical properties and formability is performed also. Since an investigation of mechanical properties of UFG materials produced by cryogenic free machining is limited, I think, that the results presented in this manuscript are interesting and original. I recommend publishing this manuscript after minor corrections given below:  

1)      The Ref. [16] is inserted into the manuscript text not on the queue.

2)      What are a method and a measurement error of chemical composition (see table 1)? Take into account that a correct result should contain one decimal digit after comma (not two) in a case of SEM/EDS results.

3)      Please correct the number of the table 4 into 2 (line 91).

4)      Сheck the name of the TEM microscope (line 108).

5)       In my opinion the sentence “Kanai et al….”(lines 201-206) is too complicated for understanding. Please change it.  

Author Response

Dear Editor and reviewers:

Thanks very much for your help and the reviewers’ comments concerning our manuscript which entitled “Preparation of ultrafine-grained continuous chips by cryogenic large strain machining”. Those comments are all valuable and helpful for revising and improving paper, as well as the significant guiding to our researchers. We have studied comments carefully and made revisions, these revisions are highlighted through the whole manuscript. Also, we have answered reviewers’ comments and explained the detail revisions in manuscript point by point. Now, we hope our revisions can meet with approval.

Response to reviewers’ comments

The authors study the influence of various machining parameters on a microstructure evolution and microhardness of the ultrafine grained Al 7075 alloy, processed by orthogonal free machining technique at room temperature (RT-FM ) and at cryogenic temperature (CT-FM). The analysis of advantages of CT-FM relative to RT-FM via mechanical properties and formability is performed also. Since an investigation of mechanical properties of UFG materials produced by cryogenic free machining is limited, I think, that the results presented in this manuscript are interesting and original. I recommend publishing this manuscript after minor corrections given below:

  1. The Ref. [16] is inserted into the manuscript text not on the queue.

Thank you so much for your comments. Your comments are very valuable to us. Due to our carelessness, we made this mistake and the order of reference is not quite correct. We changed Ref. [16] to Ref. [8], and the rest of the references in the wrong order were corrected one by one. The line numbers of the modified contents are from 44 to 66.

Based on your valuable comments, we have modified the manuscript as following:

The shear strain () imposed on shear plane is given by [8]:

Over the past decade, a lot of scholars applied machining to produce UFG materials. Swaminathan et al [9] elaborated plane strain machining is a low-cost and mass production way for making NC and UFG materials. Deng et al [10] successfully prepared UFG material with an average size of 200nm by machining carbon steel. However, SPD imposed on Al 7XXX series alloy at RT get bad deformability due to the formation of solute clusters and metastable particles, they result in samples cracking [11]. A series of scholars successive put forward solutions such as processing at elevated temperature or pre-aging treatments before SPD [3,12], but these methods lead to either material softening or formation of coarse and stable incoherent  phase by over aging. In turn, the strength of the Al 7XXX series alloys decreases. Moreover, the elevated temperature which occurred during SPD at ambient temperature causes material dynamic recovery, recrystallization and grain growth which sacrifice material mechanical strength [2,13]. Therefore, cryogenic temperature (CT) SPD is developed to conquer the aforementioned limitation. Yu et al [14] found the ultimate tensile stress of Al/Ti/Al laminate sheets processed by cryogenic roll bonding was 36.7 higher than that by room-temperature roll bonding. By the way, edge cracks also appear in the latter but the former is in good shape. Panigrahi et al [15] concluded that produce high strength UFG Al 7075 alloys with high angle grain boundaries can be achieved under cryorolling with a strain of 3.4. Shi et al [16] employed cryogenic rolling to suppress dynamic recovery and accumulation high-density dislocation in Al 5052 alloy which simultaneously raised yield strength and tensile strength of the alloy.

Now we think readers can better understand our paper, please consider our revisions.

  1. What are a method and a measurement error of chemical composition (see table 1) Take into account that a correct result should contain one decimal digit after comma (not two) in a case of SEM/EDS results.

Thank you so much for your comments. Your comments are very valuable to us. The chemical composition of Al 7075 alloys is referenced from data in published paper (doi:10.1016/j.msea.2019.138106). Owing to Al 7075 alloys’ exceptional mechanical properties, they have been applied to all walks of life. Many researchers have carried out extensive investigations on 7075, so its chemical composition has reached a consensus. We ensure the chemical composition of Al 7075 in our paper is reliable. We changed Al elements’ chemical composition to Balance, the line number of the modified contents is 89.

Based on your valuable comments, we have modified the manuscript as following:

Table 1. Chemical compositions of the Aluminum 7075 alloy (in %wt).

Al

Zn

Mg

Cu

Fe

Si

Mn

Cr

Ti

Balance

5.60

2.50

1.60

0.50

0.40

0.30

0.23

0.20

Now we think readers can better understand our paper, please consider our revisions.

  1. Please correct the number of the table 4 into 2 (line 91).

Thank you so much for your comments. Your comments are very valuable to us. Due to our carelessness, we made this mistake. We changed Table 4 to Table 2, The line number of the modified contents is 96.

Based on your valuable comments, we have modified the manuscript as following:

The machining parameters are shown in Table 2.

Now we think readers can better understand our paper, please consider our revisions.

  1. Сheck the name of the TEM microscope (line 108).

Thank you so much for your comments. Your comments are very valuable to us. After our verification, we found previous TEM name is wrong and we correct the right name. The right TEM microscope’s name is JEM-1400 PLUS, the line number of the modified contents is 113.

Based on your valuable comments, we have modified the manuscript as following:

JEM-1400 PLUS transmission electron microscopy (TEM) which operates at an acceleration voltage of 200kv was used to characterize the microstructure and precipitation of the UFG chips.

Now we think readers can better understand our paper, please consider our revisions.

  1. In my opinion the sentence “Kanai et al….”(lines 201-206) is too complicated for understanding. Please change it.

Thank you so much for your comments. Your comments are very valuable to us. In this sentence we want to convey that under low machining velocity, the actual shear strain and the generated heat don’t have a considerable value, the effect of CT is relatively small, thus the hardness difference between CT chip and RT chip is relatively small. We have adjusted the structure of this sentence, the line numbers of the modified contents are from 205 to 210.

Based on your valuable comments, we have modified the manuscript as following:

Kanani et al [21] revealed that if machining velocities are lower than threshold value which converts the chip forming mechanism, the actual shear strain imposed on the chip is smaller than the strain calculated by Eq. (1). Moreover, the generated heat at low machining velocities might not have a considerable magnitude. The factor that affects chip hardness at low speed is low strain and a trace amount of heat, thus the difference between the hardness of CT chips and the hardness of normal RT chips is relatively small.

Now we think readers can better understand our paper, please consider our revisions.
